# Individualized Analysis of Lateral Asymmetry Using Hip-Knee Angular Measures in Soccer Players: A New Methodological Perspective of Assessment for Lower Limb Asymmetry

**DOI:** 10.3390/ijerph19084672

**Published:** 2022-04-13

**Authors:** Oscar García-García, Ángela Molina-Cárdenas, Tania Álvarez-Yates, Mario Iglesias-Caamaño, Virginia Serrano-Gómez

**Affiliations:** 1Laboratory of Sports Performance, Physical Condition and Wellness, Faculty of Education and Sports, University of Vigo, 36310 Pontevedra, Spain; tanalvarez@uvigo.es (T.Á.-Y.); mariglesias@uvigo.es (M.I.-C.); vserrano@uvigo.es (V.S.-G.); 2University of Málaga, 29016 Málaga, Spain; angela.molina.cardenas@gmail.com

**Keywords:** interlimb asymmetry, ROM, soccer, gender

## Abstract

This study aimed to: (1) determine the magnitude and direction of lateral asymmetry in well-trained soccer players using hip and knee ROM tests; (2) inquire if asymmetry relies on the ROM test performed and/or gender; and (3) establish asymmetry thresholds for each ROM test to individualize lower-limbs asymmetry. One hundred amateur soccer players were assessed using hip–knee ROM tests: Straight Leg Raise, modified Thomas Test, hip internal rotation and external rotation, hip abduction (ABD) and adduction (ADD), Nachlas Test and Rigde Test. There are significant differences between tests when determining the magnitude of lateral asymmetry (F = 3.451; *p* = 0.001; η_p_^2^ = 0.031) without significant differences between gender (F = 0.204; *p* = 0.651; η_p_^2^ = 0.001). Asymmetry threshold results differ significantly between using a fixed or a specific threshold (F = 65.966; *p* = 0.001; η_p_^2^ = 0.985). All tests indicate that the direction of asymmetry is towards the dominant limb. In conclusion, the ROM test used determines the magnitude and direction of the lateral asymmetry of the amateur soccer players. The ABD and ADD are the ROM tests that showed higher percentages of asymmetry, without differences between female and male soccer players. Using a specific asymmetry threshold formula can classify more players as asymmetrical than with a fixed threshold.

## 1. Introduction

As consequence of soccer players lateral dominance, lateral asymmetries can develop, understood as differences in strength, power, stiffness or range of movement (ROM) between muscles of both sides of the body [1]. Besides, the characteristics high-intensity actions of soccer appear in most cases unilaterally [2,3], which could favor the development of these asymmetries. These typical actions with unilateral prevalence (i.e., changes of direction, jumps, kicks, etc.) can cause soccer specific adaptions that can entail an overload of a certain structure [4,5]. Different authors have addressed lateral asymmetries, referring to: dominant leg (DL) or non-dominant leg (NDL) [6], strong leg or weak leg [7], or just right and left leg [8]. Although, all report the percentage difference between limbs.

However, there is no consensus on whether the athlete’s performance is related to a greater or lesser asymmetry [9]. Bishop et al. [10] pointed out that interlimb differences in strength can decrease jumping, hitting, or even pedaling performance. However, the evidence of these interlimb asymmetries through jumping ability is still inconclusive. This is probably due to the different methods to survey asymmetry (i.e., vertical jump, strength, multidirectional speeds, etc.), assess sport performance, establish asymmetry thresholds, or even athletes’ gender. Hence, it seems to be an important complexity when it comes to approach asymmetry and to determine its relationship with physical and sports performance [10].

Soccer players are familiar with this situation, since interlimb asymmetries have already been noticed in parameters such as impulse and maximal power in countermovement (CMJ) [5], or ground reaction forces in Deep Squat [8]. These interlimb asymmetries can be moderately decreased by both bilateral and unilateral strength training [11], although it seems that exercising limbs unilaterally has greater improvements [3].

On the one hand, it has been suggested that asymmetries in single-leg CMJ were associated with lower performance in 5, 10 and 20 m sprint times, both in elite youth [12] and adult female soccer players [13]. Although, on the other hand, no relationships were established between lateral asymmetry and performance in young male soccer players assessed through an Abalakov jump, changes of direction (COD), isoinercial power and linear sprint tests [14], nor between multidirectional speed and unilateral jumping performance (vertical, standing broad and lateral jumps) [15].

Notwithstanding, multiple authors agree and point out that an interlimb asymmetry is a risk factor for suffering a sport injury [16]. In fact, a 10% side-to-side difference between limbs is considered a useful tool for detecting players with high risk of injury [17]. Previously, it has been shown that athletes with a >15% imbalance in knee flexors and/or hip extensor were at higher risk of suffering an injury [18]. More recently, Read et al. [19] pointed out that lateral asymmetries in single leg CMJ peak landing vertical ground reaction force are associated with higher risk of lower-limb injury. In addition, gender seems to be a differentiating factor, since female athletes present a greater association between lateral asymmetry and ACL injury risk [6,20].

Additionally, the existence of a contralateral deficit has been established when the asymmetry exceeds a 10–15% difference between limbs [21]. However, these thresholds seem arbitrary since it cannot be applied to all tests nor populations since it would not represent the real asymmetry of the athlete [2]. Currently, it has been suggested that applying an individual analysis based on asymmetry thresholds could be more adequate for identifying asymmetrical athletes [22]. Bishop et al. [2] pointed out that asymmetry must be individualized and verified that its percentage value is greater than the coefficient of variation (CV) of the test. Determining asymmetry through individualized thresholds would allow coaches to have reference data for specific population regarding certain tests.

Traditionally, interlimb asymmetry assessments have been carried out based on unilateral jumping (CMJ), strength (isokinetic) or COD speed. Nevertheless, hamstrings are the main muscle involved in knee and hip flexion, actions that are frequently performed by soccer players during their characterized high-speed sprinting and kicking [23]. It has been pointed out that athletes with greater flexibility traditionally present improved proficiency in movements [24] and can encourage speed performance improvements [25], while low levels have even been related to a risk for hamstring injury [26]. Specifically, it has been revealed that female and male soccer players who suffered a hamstring strain injury had lower ROM in active and passive straight leg raise, Nachlas test and Ridge tests, while a higher ROM in the Thomas test than non-injured players [26]. In fact, these authors highlighted that hip–knee ROM test are good predictors of hamstring strain injury. Yet, to our knowledge, despite the relevance that hip and knee ROM seems to have in soccer players, ROM interlimb asymmetry analysis has been little addressed, even though flexibility asymmetry is an important internal risk factor for knee injuries [24].

Hence, based on the relevance that lateral asymmetry and hip and knee ROM seems to have on soccer players’ performance and on their risk of injury, we consider it appropriate to explore lateral asymmetry using hip and knee ROM values to establish individualized reference values for female and male soccer players. We hypothesize that the type of test used, the player’s gender and the asymmetry threshold used will determine the magnitude and direction of soccer players’ ROM asymmetry. Therefore, the aim of this study was to (1) determine the magnitude and direction of lateral asymmetry in well-trained soccer players using hip and knee ROM tests; (2) inquire if asymmetry relies on the ROM test performed and/or gender, and (3) establish asymmetry thresholds for each ROM test to individualized lower limb asymmetry in soccer players.

## 2. Materials and Methods

### 2.1. Study Design

A comparative and cross-sectional study design has been used, following an associative strategy to determine the magnitude and direction of lateral asymmetry in hip–knee ROM of female and male amateur soccer players.

The ROM testing battery was performed within the players’ competitive period. Eight angular tests were carried out on both lower limbs of the soccer players to assess their hip and knee ROM: straight leg raise (SLR), modified Thomas Test (TT), hip internal rotation (IR) and external rotation (ER), hip abduction (ABD) and adduction (ADD), Nachlas Test (NT), and Rigde Test (RT). Measurements were made using the free ROM^©^ goniometric application (v.1.4) for the Samsung Galaxy S7 android smartphone, which works as a digital inclinometer allowing to record ROM in real time. These smartphone app measurements have been shown to be as reliable as those of a universal goniometer (r ≥ 0.93; ICC ≥ 0.93) [27]. The smartphone was placed 10 cm below the joint axis and held in place by the evaluator during the movement from the initial position to the final position [28]. All tests were carried out by the same experienced researcher in the use of goniometry for mobility assessments. Three attempts were made, retaining for further analysis the best ROM value performed without compensatory movements. To determine the magnitude and the direction of lateral asymmetry, the formula (1) proposed by Bishop et al. [2] was used: [DL − NDL/DL × 100] × IF (DL < NDL,1,−1).(1)

This formula implements the excel IF function that allows to monitor the direction of lateral asymmetry without magnitude variation issues (valued in %). Lateral dominance was determined based on the kicking leg of each player (DL vs. NDL). To assess data reliability, three measurements, with a 15 min interval, were taken from 15 randomly chosen players. All the proposed tests have been shown to have a high intra-day reliability (ICC = 0.96–0.99; CV 0.6–2.9%) [26].

### 2.2. Participants

A total of 100 amateur soccer players, 56 men (average 20.38 ± 3.90 years, body mass 69.06 ± 8.3 kg, height 1.76 ± 0.07 m) and 44 women (average 20.86 ± 3.46 years, body mass 61.14 ± 6.33 kg, height 1.65 ± 0.07 m) volunteer to the study. Seventy-one of them (71%) presented right-leg dominance and twenty-nine (29%) had left-leg dominance. Both male and female athletes played in the Spanish soccer league, belonging to the 3rd and 2nd National division, respectively, and trained on average three times/week. All soccer players signed an informed consent before testing and presented a healthy state, without symptoms of illness or injuries. The research protocol followed the principles of the Declaration of Helsinki regarding biomedical research involving human subjects (64th WMA General Assembly, Fortaleza, Brazil, 2013). Approval was granted by the soccer teams management boards and coaching staffs and by the Local Ethical Research Committee.

### 2.3. Procedure

All tests were performed both active (without no external manipulation from the evaluator) and passive (with external manipulation from the evaluator), except TT, NT and RT. The assessments were always performed on Mondays’ recovery session after playing the weekend game. All tests were carried out prior to training sessions and always at the same time and following the same order: first active and then passive. The assessment protocol followed Molina-Cárdenas et al. [26] ROM testing procedure.

The SLR was performed following the Ayala et al. [29] protocol for evaluating hip ROM and hamstring flexibility. The player laid in the supine position on a stretcher with arms straight at the sides and raised one single leg as far as possible without flexing the non-assessed leg. The smartphone was placed below the greater trochanter to record hip flexion.

The modified TT was used to assess the flexibility of the hip flexor muscles, especially the psoas major. The players in a supine position with knees bent over the edge of the stretcher were instructed to flex one knee to the chest and hold it following Peeler and Anderson [30] protocol. ROM was recorded maintaining the SLR smartphone position.

The IR and ER were used as tests to assess hip ROM following Sadeghisani et al. [31] sitting protocol. The player sat on the edge of a stretcher with a 90º hip and knee flexion to perform an internal and external rotation, with the smartphone placed below the inferior pole of the patella, until maximum hip ROM or any compensatory movements were observed.

The ABD and ADD were used as tests to assess hip ROM on the frontal plane. The player lied in a supine position on a stretcher and performed an abduction and adduction with their leg straightened [32]. In this test, the smartphone was placed below coxofemoral joint, over the femur.

The NT and RT were used to evaluate the quadriceps flexibility and the knee ROM. For both tests, the player lied in a prone position (for RT with a hip extension) on stretcher and flexed the knee taking the foot heel to the gluteus until lumbar spine or hip compensations begin to appear [33]. The ROM was recorded with the smartphone placed below the fibular head.

### 2.4. Statistical Analyses

Values are reported as mean ± standard deviation (SD). Sample size was calculated (using G*Power Version 3.1.9.4), introducing the following parameters: effect size (ES) 0.55 and α error probability (0.05) and power (0.95), which resulted in a sample size of 90 participants. A paired simple T-Test was used to compare ROM between the DL and NDL.

The influence on the magnitude and orientation (DL vs. NDL) of the asymmetry as a function of the tests (within-subjects) and gender (between-subjects) factors were assessed with a two-factor ANOVA after previously assuming multivariate normality and homogeneity of variances and covariances. Application of the univariate Kolmogorov–Smirnov test, in conjunction with the Lilliefors test, showed that the sample distribution was normal and linear. Homoscedastic assumption was verified with the Box M test followed by a post hoc HSD Turkey test. Finally, to classify a player as “asymmetrical” in a ROM test, two procedures were carried out: fixed traditional asymmetry thresholds (10–15% difference between legs) [7,21], and specific asymmetry thresholds following Dos’Santos et al.’s [22] (formula (2)) where % Asym and SD are the average percentage and standard deviation of the sample’s asymmetry, respectively:% Asym + (0.2 × SD) (2)

Additionally, a two-way ANOVA was used to determine if the number of players classified as “asymmetrical” were modulated by the asymmetry threshold used and/or by gender. The effect sizes in ANOVA two-way were reported as partial eta square (η_p_^2^) and interpreted as small (0.01), moderate (0.06), or large (0.14) [34]. An alpha level of *p* < 0.05 was considered statistically significant. All data were analyzed using SPSS v.24.0 for Windows (SPSS Inc., Chicago, IL, USA).

## 3. Results

Players’ NDL has greater ROM than DL in ABDa (*p* = 0.003); ABDp (*p* = 0.004) and RT (*p* = 0.001). No differences were found in the rest of the ROM tests.

The results of the analysis of variance indicate that there are significant differences between tests when determining the magnitude of lateral asymmetry (F = 3.451; *p* = 0.001; η_p_^2^ = 0.031) with a small effect size. On the contrary, no difference was found between female and male players (F = 0.204; *p* = 0.651; η_p_^2^ = 0.001) nor in *test* × *gender* interaction.

Table 1 shows the magnitude of lateral asymmetry obtained by soccer players in each test according to gender.

If the percentage of asymmetry is calculated with the ABDa tests it is much higher than with SLRa (*p* = 0.001), SLRp (*p* = 0.001), TT (*p* = 0.001), IRa (*p* = 0.002), IRp (*p* = 0.001), ERa (*p* = 0.001), ERp (*p* = 0.001), ADDp (*p* = 0.001), NT (*p* = 0.021) and RT (*p* = 0.003). Similarly, if it is calculated with the ABDp test it is much higher than with SLRa (*p* = 0.037), SLRp (*p* = 0.005), TT (*p* = 0.014), ERp (*p* = 0.022) and ADDp (*p* = 0.044). Additionally, if it is calculated with the ADDa test it is greater than with SLRa (*p* = 0.009), SLRp (*p =* 0.001), TT (*p* = 0.003), IRp (*p* = 0.021), ERa (*p* = 0.030), ERa (*p* = 0.005), and ADDp (*p* = 0.011). Finally, if calculated with NT, it is higher than with SLRp (*p* = 0.026).

The magnitude in each test when the direction of lateral asymmetry is added again indicates significant differences between tests when determining the magnitude and direction of lateral asymmetry (F = 2.167; *p =* 0.011; η_p_^2^ = 0.020) with a small effect size. There are no differences between male and female players (F = 0.975; *p* = 0.324; η_p_^2^ = 0.001), nor in the *test* × *gender* interaction (F = 1.012; *p* = 0.435; η_p_^2^ = 0.009). Except in the TT (0.3509%), all tests indicate that the direction of asymmetry is towards the DL (a negative value), that is, the NDL presents more flexibility. However, it must be pointed out that greater values achieved in TT (more degrees) represents a worse hip mobility, which in other words, shows that the NDL obtains again greater results than the DL. The greatest asymmetry towards the DL is obtained in the ADDa test (−1.71717%) and in the ABDa (−1.49186%).

The individualized analysis for each player based on each ROM test asymmetry threshold using (a) the specific asymmetry threshold formula or (b) the 10% fixed asymmetry points out that there are significant differences between both thresholds to determine soccer players’ lateral asymmetry (F = 65.966; *p =* 0.001; η_p_^2^ = 0.985) with a large effect size. In addition, the *threshold* × *gender* interaction (F = 4.070; *p* = 0.049; η_p_^2^ = 0.078) also shows a moderate effect size. However, there are no differences in gender (F = 2.999; *p* = 0.333; η_p_^2^ = 0.750). As it can be observed in Table 2 when using the specific thresholds formula, the percentage of classified players as asymmetrical is between 16% for the ADDa test and 39% of the ABDa test. However, if the fixed threshold of 10% difference between limbs is used, the percentage of asymmetrical players significantly decreases between 0% and 6% (ADDa test). The ABDa (39.2%), SLRa (35.7%) and TT (33.9%) are the ROM tests with the highest percentage of asymmetrical male players, while the ERp (47.7%), ERa (31.8%) and ABDp (36.3%) are the ROM tests that classify a higher percentage of female players as asymmetrical.

## 4. Discussion

The main findings indicate that the applied ROM test determines the magnitude and direction of the lateral asymmetry of the amateur soccer players. The ABDa, ABDp and ADDa are the ROM tests in which higher percentages of asymmetry are obtained, without differences between females and male players. Players’ NDL has greater ROM than DL in ABDa, ABDp and RT, without differences between limbs in the rest of ROM tests. The individualized analysis of the lateral asymmetry for each player shows that the used threshold modulates the percentage of asymmetrical players in the ROM tests. When using the fixed 10% interlimb difference threshold, the percentage of asymmetrical players decreases notably with respect to percentage of asymmetric players classified with a specific asymmetry threshold formula.

Our findings indicate that the type of test used will modify the result obtained. If we also consider the sports discipline, age, level of performance, etc., it is necessary to be very careful when comparing the ROM values obtained in this work with those from other authors. For instance, there is some reports regarding passive SLR ROM tests. Our ROM values in this test are slightly higher than those found by found by López-Valenciano et al. [34] in male professional soccer players (85.7° vs. 80.7°), which were also greater that those obtained with futsal players (85.8° vs. 72.3° left and 85.5° vs. 77.4° in right limb, 89.9° vs. 78.1° left and 89.7° vs. 78° right limb in male and female players, respectively) [29] and with male 1st Division handball players (85.6° vs. 78.5°). Yet, male 2nd Division handball players values were similar to ours (85.6° vs. 86.3°) [32]. On the contrary, our average ROM values in IRp (38.8° vs. 46.2°) and ERp (48.4° vs. 50.3°) are lower than those found by López-Valenciano et al. [35] in male professional soccer players. However, it should be noted that these ROM values are modified during the season, since progressive reduction in hip extension ROM has been already observed throughout soccer players season [36]. Therefore, the assessment moment withing the season could have also influenced our results.

As abovementioned, the type of ROM test used seems to be very relevant, since our results indicate that the type of test used determines the magnitude and direction of lateral asymmetry. These findings are in line with those of Bishop et al. [37] when evaluating asymmetry through strength tests. Hence, it seems necessary to establish which are the most appropriate ROM tests for each sport. However, the magnitude and direction of lateral asymmetry for a given test cannot be applicable to all ROM tests. That is, a soccer player may be not considered asymmetrical in a certain ROM test, while could be asymmetrical in another one, being perfectly compatible.

Our results show that NDL has more ROM in ABDa, ABDp and RT, without differences in the rest of the tests both passively and actively (SLR, TT, IR, ER, ADD and NT). In this line, no significant differences have been found neither between the DL and NDL in passive ER and IR in un-injured Australian professional soccer players [38]. Nor have López-Valenciano et al. [35] found, like us, significant differences between DL and NDL in none of the hip ROM test carried out in professional soccer players, except in passive ABD test that we have found significant differences between DL and NDL. On the contrary, it has been suggested that the DL had greater hip joint flexibility respect to NDL in youth elite soccer players in the passive SLR [24]. As ROM values, inter-limbs asymmetries can also change during the season, since it has been observed that hip IR ROM increased from soccer players’ pre-season to mid-season [36]. Hence, the moment to assess inter-limbs asymmetry could also influence the players results. Based on our results, soccer player’s hip–knee ROM shows a homogenous direction of lateral asymmetry towards the DL. In other words, soccer players have more ROM in the NDL, which may be reasonable explained due to soccer’s sport demands, specifically in the actions that are performed to a greater extent with DL such as dribbling, kicking, shooting, etc. Soccer players use one favored limb unilaterally for kicking the ball [39], which produces a habitual use of dynamic stretching and would cause greater flexibility in the DL [24]. However, this could be interpreted contrary to these latter authors since it also implies a muscle damage, that without a proper recovery could lead to ROM and strength reductions by altering the mechanical and muscle-tendon properties of the muscular structures involved [40]. Therefore, the NDL would not normally suffer these strong concentric and eccentric contractions at shortened contracted positions, which would produce less muscle damage and more ROM than DL. Nevertheless, there is still not enough evidence, so this should still be deepened in for future research.

Taking all these findings together, it would not be possible to classify a player as symmetrical or asymmetrical with certainty. In fact, the limitations that have been highlighted for interpreting the magnitude and direction of lateral asymmetry analyzed together suggests that an individual approach for each player is necessary to determine if there is a lateral asymmetry. In the scientific literature, it is commonly used the interlimb fixed percentage (>10–15%) to indicate a high probability of injury [17,18]. However, other authors established that ROM interlimb differences >6° was an indicative of lateral asymmetry [35,41].

However, the threshold for considering an athlete as asymmetrical should be established based on the sample and the type of test [2]. In fact, it has been suggested that the commonly accepted 10% threshold for classifying individuals as asymmetrical should be reconsidered and reinvestigated [42].

Our findings indicate that if ROM asymmetry is classified based on a specific threshold formula there will be many more asymmetrical players without differences between females and males (over 16% and 39% of players in ADDa and ABDa, respectively) than if the 10% fixed threshold is applied (over 0–6% of players) in all ROM tests. For example, 31% of the players of our sample would be classified as asymmetrical in active hip external rotation (ERa), while using the 10% fixed threshold, no player would be considered as asymmetrical. The same happens in the rest of ROM tests. Dos’ Santos et al. [43] have obtained similar findings, classifying 13 of 43 athletes (30%) as asymmetrical using the specific thresholds formula in a 505 agility test, while using the >10% fixed threshold only 2 (5%) were identified. These findings seem to reinforce the need to assess bilateral asymmetry from an individual perspective, determining the asymmetry threshold based on the test and the group of players.

To our knowledge, few authors have determined the percentage of interlimb flexibility asymmetry in soccer players. It has been reported that the percentage of asymmetrical players in terms of hamstring flexibility was 38% in 3rd division, establishing a cut-off of >6° [40], and with the same cut-off, 30% of Spanish professional soccer players presented bilateral asymmetry in hip, knee or ankle ROM measures [35]. These results are in line with the asymmetry results obtained in our male amateur players using “specific thresholds formula” (between 20% in IRp and 39% in ABDa).

Nevertheless, asymmetry values could be seriously affected by the athlete’s fitness, at least at recreational level [44]. Theses authors pointed out that asymmetries could be a consequence of a low training level and motor competence to perform a certain task. Our results are obtained from experienced soccer players who trained 3 sessions/week which carry out flexibility exercises, although they take up little time per session, but are common. Hence, the asymmetry values of hip and knee ROM could be better approached from the sporting asymmetry perspective proposed by Maloney [9]. That is, the magnitude of ROM lateral asymmetry is caused by the sports practice itself. In fact, this is not new, since it has already been suggested that soccer players’ lower limbs tend to be asymmetrical in flexibility, muscle strength concentric and eccentric, knee laxity, etc. [41]. The lateral predominance to perform many of soccer game actions (i.e., hitting, dribbling, etc.) could induce alterations in lower limbs joints ROM, which could be the cause of this ROM interlimb asymmetry [24,35]. In addition, asymmetries and strength ratio imbalances tend to be more prevalent in soccer players with short and intermediate training—age, while players with high professional training experience adopt a more symmetric use of their lower limbs [39], so it seems that soccer player’s experience would also modulate the degree of asymmetry. However, to be able support this statement, it would be necessary to assess the ROM of a control group to stablish the difference between both. Without any doubt, this is a point to focus our attention.

As Fousekis et al. [41] pointed out, there is a hypothesis that symmetrical myodynamic function of the lower extremities is very important for injury prevention. Due to the relevance that lateral asymmetry and hip–knee ROM seems to have on soccer players performance and on their risk of injury, these findings could be very useful for physical trainers to assess hip–knee ROM and classify athletes depending on the individual asymmetry. This can be very useful for establishing appropriate flexibility training strategies to avoid or reduce ROM interlimb asymmetry in female and male soccer players.

Future research lines could aim to verify the predictive capacity that ROM lateral asymmetry has on the probability of suffering a muscle or tendon injury in the musculoskeletal structures involved in these specific movement. Likewise, delve into the factors that could explain the differences found between NDL and DL in female and male soccer players.

## 5. Conclusions

The ROM test used determines the magnitude and direction of the lateral asymmetry of the amateur soccer players. The ABDa, ABDp and ADDa are the ROM tests with which higher percentages of asymmetry are obtained, without differences between females and male soccer players. Players’ NDL has greater ROM than DL in ABDa, ABDp and RT, without differences between limbs in the rest of ROM tests. Coaches and physical trainers should be especially careful in choosing ROM tests to assess lateral asymmetry when it comes to classify players as asymmetrical. Likewise, when deciding whether to apply a fixed or a specific threshold to classify, since the results vary considerably. However, based on our results, it seems more appropriate to use a specific threshold to classify asymmetric players. This will can be very useful for establishing appropriate flexibility training strategies to avoid or reduce ROM interlimb asymmetry in female and male soccer players.

## Figures and Tables

**Table 1 ijerph-19-04672-t001:** Descriptive statistics of ROM tests and the magnitude of interlimb asymmetry for each test.

Test	Gender	Mean (°) ± SD		Mean % Asymmetry ± SD	Asymmetry ICC 95%
		DL	NDL		
SLRa	Females	87.25 ± 6.772	87.32 ± 6.819	0.83 ± 1.599	0.311–1.980
	Males	82.68 ± 9.052	82.89 ± 8.858	1.59 ± 1.779	0.579–2.610
	Total	84.69 ± 8.405	84.84 ± 8.286	1.26 ± 1.735	0.449–1.980
SLRp	Females	89.70 ± 3.246	89.98 ± 3.769	0.60 ± 1.187	0.541–1.750
	Males	85.75 ± 8.352	85.63 ± 8.078	1.01 ± 1.731	0.004–2.027
	Total	87.49 ± 6.872	87.54 ± 6.866	0.83 ± 1.522	0.043–1.573
TT	Females	1.11 ± 2.264	1.23 ± 2.410	0.66 ± 1.705	0.481–1.810
	Males	1.48 ± 2.123	2.09 ± 2.539	1.34 ± 1.798	0.327–2.358
	Total	1.32 ± 2.183	1.71 ± 2.508	1.04 ± 1.781	0.238–1.769
IRa	Females	39.75 ± 2.598	39.93 ± 2.929	1.15 ± 2.279	0.005–2.296
	Males	37.89 ± 4.039	38.07 ± 3.808	2.08 ± 3.912	1.061–3.091
	Total	38.71 ± 3.585	38.89 ± 3.556	1.67 ± 3.312	0.848–2.379
IRp	Females	40.32 ± 2.639	40.36 ± 2.997	1.34 ± 2.584	0.199–2.490
	Males	38.70 ± 2.972	38.98 ± 2.895	1.42 ± 3.331	0.402–2.433
	Total	39.41 ± 2.930	39.59 ± 3.005	1.38 ± 3.011	0.616–2.146
ERa	Females	51.45 ± 3.586	51.68 ± 3.241	1.76 ± 2.297	0.611–2.902
	Males	47.68 ± 4.473	47.79 ± 4.263	1.16 ± 2.225	0.144–2.174
	Total	49.34 ± 4.500	49.50 ± 4.294	1.42 ± 2.265	0.692–2.223
ERp	Females	52.88 ± 4.144	53.00 ± 4.058	1.32 ± 1.644	0.175–2.466
	Males	48.37 ± 3.539	48.52 ± 3.352	0.88 ± 1.762	0.138–1.892
	Total	50.36 ± 4.414	50.49 ± 4.289	1.07 ± 1.717	0.333–1.864
ABDa	Females	50.89 ± 3.792	51.64 ± 4.389	3.48 ± 3.549	2.337–4.628
	Males	45.27 ± 6.363	45.95 ± 5.719	3.14 ± 3.829	2.123–4.154
	Total	47.74 ± 6.050	48.45 ± 5.882	3.29 ± 3.690 *	2.545–4.076
ABDp	Females	52.68 ± 4.247	53.86 ± 5.074	3.28 ± 3.828	2.133–4.424
	Males	47.46 ± 5.243	47.63 ± 4.804	1.45 ± 2.534	0.437–2.467
	Total	49.76 ± 5.466	50.37 ± 5.802	2.26 ± 3.281 *	1.600–3.131
ADDa	Females	28.68 ± 5.020	29.57 ± 2.106	3.18 ± 15.188	2.038–4.329
	Males	28.29 ± 3.667	28.46 ± 3.406	2.12 ± 4.269	1.110–3.140
	Total	28.46 ± 4.296	28.95 ± 2.945	2.59 ± 10.517	1.889–3.420
ADDp	Females	30.20 ± 2.906	30.32 ± 2.851	0.88 ± 2.224	0.260–2.030
	Males	29.02 ± 2.901	29.07 ± 2.776	1.62 ± 3.249	0.609–2.640
	Total	29.54 ± 2.949	29.62 ± 2.863	1.30 ± 2.854	0.489–2.020
NT	Females	138.07 ± 10.244	136.98 ± 12.793	1.19 ± 1.985	0.042–2.333
	Males	132.38 ± 10.458	129.96 ± 11.704	2.89 ± 3.849	1.875–3.906
	Total	134.88 ± 10.696	133.05 ± 12.626	2.14 ± 3.266	1.273–2.804
RT	Females	119.41 ± 7.295	120.48 ± 8.194	1.52 ± 2.173	0.375–2.666
	Males	113.54 ± 8.356	115.00 ± 8.048	1.78 ± 2.406	0.761–2.792
	Total	116.12 ± 8.396	117.41 ± 8.521	1.66 ± 2.298 *	0.883–2.414

SLRa (active Straight Leg Raise); SLRp (passive Straight Leg Raise); TT (Thomas Test); IRa (active hip Internal Rotation); IRp (passive hip Internal Rotation); ERa (active External Rotation); ERp (passive External Rotation), ABDa (active hip Abduction); ABDp (passive hip Abduction); ADDa (active Adduction), ADDp (passive Adduction), NT (Nachlas Test), RT (Rigde Test), SD (standard deviation), and ICC (intraclass coefficient correlation). * (*p* < 0.05).

**Table 2 ijerph-19-04672-t002:** Individualized magnitude of asymmetry analysis of the players for each ROM test.

Test	Gender	Asymmetric Players Based on Specific Thresholds Formula		Asymmetric Players Based on 10% Fixed Thresholds	
		N	%	N	%
SLRa	Females	10	22.73	0	0
	Males	20	35.71	0	0
	Total	27	27	0	0
SLRp	Females	11	25	0	0
	Males	12	21.43	0	0
	Total	26	26	0	0
TT	Females	8	18.18	0	0
	Males	19	33.93	0	0
	Total	25	25	0	0
IRa	Females	12	27.27	0	0
	Males	15	26.79	4	7.14
	Total	28	28	4	4
IRp	Females	12	27.27	0	0
	Males	11	19.64	4	7.14
	Total	23	23	4	4
ERa	Females	14	31.82	0	0
	Males	15	26.79	0	0
	Total	31	31	0	0
ERp	Females	21	47.73	0	0
	Males	14	25	0	0
	Total	35	35	0	0
ABDa	Females	15	34.09	1	2.27
	Males	22	39.29	2	3.57
	Total	39	39	3	3
ABDp	Females	16	36.36	2	4.55
	Males	18	32.14	1	1.79
	Total	31	31	3	3
ADDa	Females	5	11.36	3	6.82
	Males	13	23.21	3	5.36
	Total	16	16	6	6
ADDp	Females	8	18.18	1	2.27
	Males	15	26.79	2	3.57
	Total	23	23	3	3
NT	Females	9	20.45	0	0
	Males	17	30.36	5	8.93
	Total	30	30	5	5
RT	Females	12	27.27	0	0
	Males	18	32.14	1	1.79
	Total	30	30	1	1

SLRa (active Straight Leg Raise); SLRp (passive Straight Leg Raise); TT (Thomas Test); IRa (active hip Internal Rotation); IRp (passive hip Internal Rotation); ERa (active External Rotation); ERp (passive External Rotation), ABDa (active hip Abduction); ABDp (passive hip Abduction); ADDa (active Adduction), ADDp (passive Adduction), NT (Nachlas Test), RT (Rigde Test); N (number of players); % (percentage of player); and specific thresholds formula: %Asim + (0.2 × SD).

## Data Availability

The data presented in this study are available on request from the corresponding author.

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
