# Peer review of "Individualized Analysis of Lateral Asymmetry Using Hip-Knee Angular Measures in Soccer Players: A New Methodological Perspective of Assessment for Lower Limb Asymmetry"

_ijerph, 2022, doi:10.3390/ijerph19084672_

Round 1
Reviewer 1 Report
Dear Authors,
Your article is fascinating in terms of its subject and content. Thank you very much for your effort. I have read all of your research with interest, and I would like to say that I like the subject. After some verifications that I will mention below, your article is suitable for publication by me.
-Some English spellings were used incorrectly in your research (like Proceedure, Asim). There are also some grammatical errors. Please review in detail.
- Add your main hypothesis and, if any, secondary hypotheses at the end of the introduction.
- If you add the effect levels of Cohen'd Effect or partial eta to your DL and NDL comparisons in Table 1, you support the evidence you present by showing the effect levels of your results. (I leave it to you, you can not do it if you want)
-I recommend you to use regression analysis in future research. (Just a recommendation ? )
-Providing suggestions for future research in the conclusion will contribute to the readers' design of new studies.
Congratulations…
Author Response
Dear reviewer 1,
Thank you very much for your kind comments. We really do appreciate them. It is very stimulating to perceive that our work can be interesting for other colleagues, which encourages us to continue in this line of research. We really reiterate our gratitude.
Point 1: Some English spellings were used incorrectly in your research (like Proceedure, Asim). There are also some grammatical errors. Please review in detail.
Response 1: Thank you very much, we have thoroughly reviewed the entire manuscript in search of other typos.
Point 2: Add your main hypothesis and, if any, secondary hypotheses at the end of the introduction.
Response 2: As you have suggested, we have introduced a hypothesis at the end of the introduction section, just before presenting the aim of our article (L95-97).We have added the following:
“We hypothesize that the type of test used, the player's gender and the asymmetry threshold used will determine the magnitude and/or direction of the soccer player's ROM asymmetry”
Point 3: If you add the effect levels of Cohen'd Effect or partial eta to your DL and NDL comparisons in Table 1, you support the evidence you present by showing the effect levels of your results. (I leave it to you, you can not do it if you want)
Response 3: Thank for very much for your comment. It is true that presenting effect levels in Table 1 would provide extra information to support our results. Yet, Table 1 current state already has a large extension and inserting more data would enlarge it even more. However, if you consider it necessary, we can try to introduce these effect levels, or if you wish we can even provide them to you without any problem.
Point 4: I recommend you to use regression analysis in future research. (Just a recommendation ?)
Response 4: We completely agree with your suggestion. The use of a predictive model it is on what we are currently working on. In this case, using lateral asymmetry as a predictor of muscle injury. It is heartwarming for us to share this same orientation with you.
Point 5: Providing suggestions for future research in the conclusion will contribute to the readers' design of new studies.
Response 5: Following your suggestion, we have added a paragraph at the end of the discussion as follows:
“Future research lines could aim to verify the predictive capacity that ROM lateral asymmetry has on the probability of suffering a muscle or tendon injury involved in the musculoskeletal structures involved in these specific movement. Likewise, delve into the factors that could explain the differences found between NDL and DL in female and male soccer players.”
Reviewer 2 Report
Thank you for reviewing your valuable research.
I have written some reviews below that will help you in your research.
1. Introduction section
line 48
Please write the first abbreviation in full words in the manuscript.
2. Materials and Methods
2.1. Study Design
- Has one evaluator performed all the evaluations? Describe the number of evaluators who performed the evaluation.
2.2. Participants
Please describe the exclusion criteria for the study subjects.
2.3. Procedure
Please describe the smartphone attachment location and measurement method for ROM measurement.
Author Response
Dear reviewer 2,
Thank you very much for all your suggestions. We really do appreciate them. Therefore, we have carried out some changes in accordance with your recommendations, and we hope that the modified manuscript will be to your liking.
Point 1: Introduction section, line 48
Please write the first abbreviation in full words in the manuscript.
Response 1: Thank you, it was overlooked. Done.
Point 2: Materials and Methods. Study Design. Has one evaluator performed all the evaluations? Describe the number of evaluators who performed the evaluation.
Response 2: Yes, exactly. All measurement were carried out by one experienced evaluator in the use of goniometry for mobility assessments (L115). For this, the evaluator had the help of another researcher to avoid possible compensatory movements and to record data. Hence, in all the evaluations there was a minimum of two people, the main evaluator in charge of measuring and using the goniometer and the assistant.
Point 3: Participants. Please describe the exclusion criteria for the study subjects.
Response 3: Players who were injured at the time of the assessments, who were not in a healthy state or for any other reason had not trained regularly in the last month prior to data collection were excluded of the study. To clarify this in the text, we have added this information in the participant’s section as follows (L134):
“All female and male soccer players that were included in our study presented a healthy state, without symptoms of illness or injury”.
Point 4: 2.3. Procedure. Please describe the smartphone attachment location and measurement method for ROM measurement.
Response 4: The Smartphone was placed 10 cm below the joint axis and held in place by the evaluator during the movement from the initial position to the final position. This information was added to the text within the study design section (L115-116).
In addition, the specific location for each test was briefly specified in the procedure section.
Reviewer 3 Report
The following items are suggested for further revision.
- Line 110: While the use of smartphone instead of traditional goniometer, the location/position of smartphone need to be addressed in each test so as to indicate how the measurement of smartphone could be used as the same as traditional goniometer. This description may be useful to use in the discussion section for the results differences between literature findings.
- Line 128: As comparison between dominance and non-dominance is a key part of this study, the methodology about how to identify dominance side need to be addressed.
- Line 142: Check if typo error “them”
- Line 177: The full team of “%Asim” need to be remarked
- Line 178-179: the classification method for “asymmetrical” is missing.
Author Response
Dear reviewer 3,
Thank you very much for your review. We really appreciate all your suggestions for improving our manuscript. We have acted on your recommendations and trust that the amended paper is to your satisfaction.
Point 1: Line 110: While the use of smartphone instead of traditional goniometer, the location/position of smartphone need to be addressed in each test so as to indicate how the measurement of smartphone could be used as the same as traditional goniometer. This description may be useful to use in the discussion section for the results differences between literature findings.
Response 1: The Smartphone was placed 10 cm below the joint axis and held in place by the evaluator during the movement from the initial position to the final position. This information was added to the text within the study design section (L115-116).
In addition, the specific location for each test was briefly specified in the procedure section.
Point 2: Line 128: As comparison between dominance and non-dominance is a key part of this study, the methodology about how to identify dominance side need to be addressed.
Response 2: Lateral dominance was determined based on the kicking leg of each player. That is, the kicking limb was considered the dominant leg. Players were asked which leg they would use to take a penalty or any other action with the ball. This point was corroborated with the analysis of the game since the evaluators had access to the players’ usual training sessions. We have previously addressed how we have identified the dominant and non-dominant leg in the study design section (L122)
Point 3: Line 142: Check if typo error “them”
Response 3: Thank you very much, corrected
Point 4: Line 177: The full team of “%Asim” need to be remarked
Response 4: The formula is explained as follows:
% Asym: average percentage of the sample’s asymmetry
0.2: minimum detectable change
SD: standard deviation of the sample’s asymmetry
Point 5: Line 178-179: the classification method for “asymmetrical” is missing.
Response 5: To classify the players as asymmetrical, two procedures were used: the fixed asymmetry threshold and the specific asymmetry threshold using Dos'Santos et al. formula.
Using the fixed asymmetry threshold, a player was considered as asymmetrical in a ROM test, when the difference between DL and NDL exceeded 10%. Using the specific asymmetry threshold, a player was considered as asymmetrical in a ROM test when the % of difference between DL and NDL exceed the value obtained with the % Asym formula (0.2 x SD).
This information has been added to the manuscript (L176)